



# Quantifying aerosol size distributions and their temporal variability in the Southern Great Plains, USA

Peter J. Marinescu[1], Ezra J. T. Levin[1], Don Collins[2], Sonia M. Kreidenweis[1], Susan C. van den Heever[1]

[1]Department of Atmospheric Science, Colorado State University, Fort Collins, 80526, USA
[2]Department of Chemical and Environmental Engineering, University of California, Riverside, 92521, USA

*Correspondence to*: Peter J. Marinescu (peter.marinescu@colostate.edu)

**Abstract.** A quality-controlled, 5-year dataset (2009-2013) of aerosol number size distributions (particles with diameters ($D_p$) from 7 nm through 14 μm) was developed using observations from a scanning mobility particle sizer, aerodynamic particle sizer, and a condensation particle counter at the Department of Energy's Southern Great Plains (SGP) site. This

dataset was used for two purposes. First, typical characteristics of the aerosol size distribution (number, surface area, and volume) were calculated for the SGP site, both for the entire dataset and on a seasonal basis, and size distribution lognormal fit parameters are provided. While the median size distributions generally had similar shapes (4 lognormal modes) in all the seasons, there were some significant differences between seasons. These differences were most significant in the smallest particles ($D_p$<30nm) and largest particles ($D_p$>800nm). Second, power spectral analysis was conducted on this long-term

dataset to determine key temporal cycles of total aerosol concentrations, as well as aerosol concentrations in specified size ranges. The strongest cyclic signal was associated with a diurnal cycle in total aerosol number concentrations that was driven by the number concentrations of the smallest particles ($D_p$<30nm). This diurnal cycle in the smallest particles occurred in all seasons, in ~50% of the observations, suggesting a persistence influence of new particle formation events on the number concentrations observed at SGP. This finding contrasts with earlier studies that suggested new particle formation is observed

primarily in the springtime at this site. The timing of peak concentrations associated with this diurnal cycle was shifted by several hours depending on the season, which was consistent with seasonal differences in insolation and boundary layer processes. Significant diurnal cycles in number concentrations were also found for particles with $D_p$ between 140 nm and 800 nm, with peak concentrations occurring in the overnight hours, which were primarily associated with both nitrate and organic aerosol cycles. Weaker cyclic signals were observed for longer time scales (days to weeks) and are hypothesized to

be related to the time scales of synoptic weather variability. The strongest periodic signals (3.5-5-day and 7-day cycles) for these longer time scales varied depending on the season, with no cyclic signals and the lowest variability in the summer.

## 1 Introduction

Aerosol particles play a number of roles in the Earth-Atmosphere system, including impacting warm and cold cloud formation, solar and terrestrial radiation budgets, and human and environmental health. These impacts depend strongly on





particle size, composition, and abundance. Aerosol number and mass concentrations arise from numerous sources and processes, including in situ chemical conversion, that shape the resulting chemical compositions and size distributions of the particle populations. Long-term observations provide insights to these processes by creating datasets that enable robust statistics regarding the typical temporal variations in aerosol properties. One such site with long-term aerosol measurements

is the United States Department of Energy's Atmospheric Radiation Measurement's Southern Great Plains (SGP) site. Located in north central Oklahoma, the ARM-SGP site (Sisterson et al., 2016) is influenced by a variety of aerosol types, sources, and transport pathways (e.g., Peppler et al., 2000; Sheridan et al., 2001; Andrews et al., 2011), making it an ideal location to study a wide range of aerosol processes and to characterize aerosol properties for a typical North American, rural, continental site.

Several studies have utilized the long-term aerosol data at the SGP site to study aerosol temporal variability. Sheridan et al. (2001) provided a climatology using 4 years of data of aerosol optical properties at SGP, as well as monthly, daily, and hourly statistics of total aerosol number concentrations for particles with diameters ($D_p$) between ~10 nm and 3 µm. They found a diurnal cycle in total aerosol number concentrations that reached a minimum between 09 and 16 UTC, equivalent to 04 and 11 Central Daylight Time (CDT; CDT = UTC-5), and reached a maximum between 19 and 22 UTC (14

and 17 CDT). They also found a weak weekly cycle in aerosol number concentrations, with minimum concentrations on Sunday. However, their study did not assess the diurnal or weekly variability on a seasonal basis. Most recently, Sherman et al. (2015) assessed the temporal variability of aerosol optical properties at 4 different sites in the United States, including SGP. They found that aerosol optical properties (e.g., scattering and absorption coefficients of aerosol with $D_p < 1$ µm) had higher amplitude variations associated with seasonal time scales than with weekly or diurnal timescales at the individual

sites, and that the seasonal variations at individual sites were larger than regional variations for the same season. Both findings support the need to understand aerosol processes on a seasonal basis. Sherman et al. (2015) was a follow-up study to, and generally consistent with, the results of Delene and Ogren (2002) and Sheridan et al. (2001), with all three studies focusing on aerosol optical properties at the SGP site. These studies demonstrated weak diurnal and weekly cycles of aerosol scattering and absorption that were significant depending on the season, with absorption having a stronger signal. Parworth

et al. (2015) also provided some evidence of diurnal cycles in aerosol properties at the SGP site using 18 months of speciated aerosol mass concentration data ($D_p$ between 100 nm and 1 µm). Jefferson et al. (2017) related some of the results from these prior studies to the seasonal variability in aerosol scattering coefficient hygroscopic growth with 7 years of SGP data.

None of these prior studies of long-term variability in aerosol properties at the SGP site exploited the multiyear datasets of number size distributions available for the site, which allow for specific size ranges of aerosol particles to be

studied. Number size distributions have been used to understand a variety of aerosol processes, such as new particle formation and growth (e.g., Dal Maso et al., 2005; Hallar et al., 2011; Pierce et al., 2014; Yu et al., 2015; Niemenen et al., 2018) and cloud processing of aerosol size distributions (e.g., Weingartner et al., 1999), at long-term aerosol observing sites around the world. Here, we present and analyze 5 years of aerosol number size distribution data ($D_p$ between 7 nm and 14 µm) from the SGP site. Specifically, we develop descriptions of annually and seasonally averaged sub- and super-micron

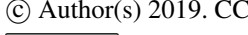


size distributions and quantify their variability. Such descriptions are useful for validating aerosol models on a variety of scales, and for selecting aerosol properties representative of the SGP site and the region. Representative aerosol size distributions at SGP are especially important for guiding the characteristics, location, and life cycle of aerosol particles in numerical modelling studies that try to represent the impacts of aerosol particles on the Earth system (e.g., Fridlind et al.,

2017; Marinescu et al., 2017; Saleeby et al., 2016). Further, the long-term time series contain information on temporal cycles that can lead to insights into the aerosol sources and processes at SGP. In this work, we apply power spectral analysis to the time series of aerosol size distributions to determine the presence of significant temporal cycles in the aerosol data.

## 2 Data

The data presented here were collected at the SGP central facility (lat = 36.605, lon = -97.485), representing a

typical North American, rural, continental site. This site has many atmospheric science observations platforms, all located within an approximately 1 km$^2$ area (Sisterson et al., 2016). This site is located within a large agricultural region in the central United States, which grows a variety of crops such as winter wheat, soybeans, cotton, corn and alfalfa and has open pasture land (USDA-NASS Oklahoma Field Office). Therefore, agricultural aerosol sources frequently impact the aerosol conditions observed at the SGP site. There are a few local power plants (e.g., a coal-fired power plant in Red Rock,

Oklahoma, 30 km to the southeast) and oil refineries (e.g., near Ponca City, Oklahoma, 35 km to the east), and Oklahoma City is approximately 130 km to the south. Besides local sources, the SGP site often encounters large concentrations of aerosol particles via long-range transport. High concentrations of aerosol particles associated with biomass burning in Central America and Mexico have been well documented in the spring and summer months (e.g., Peppler et al. 2000; Sheridan et al. 2001), although localized agricultural burning is also present (e.g., Parworth et al. 2015). Dust aerosol

particles from both local sources and long-range transport have been observed at the SGP, as well (e.g., Andrews et al. 2011).

A scanning mobility particle sizer (SMPS), which was part of the tandem differential mobility analyzer system (TDMA), measured particle size distributions between approximately 12 and 750 nm (Collins 2010) during the 2009-2013 period at the SGP site. The size distributions were typically measured in 42-49-minute time intervals, which was longer than

typical SMPS measurements due to simultaneous operation of the instrument as a TDMA to measure aerosol hygroscopicity. In this study, the data were binned into 2-hour intervals to create a more robust and evenly spaced dataset for analysis. For most of this time period, observations from an aerodynamic particle sizer (APS; TSI model 3321) were combined with the SMPS data to construct a number size distribution from ~12 nm to ~14 μm with 215 size bins (SMPS+APS; ARM Climate Research Facility, 2010, 2015). An assumed particle density of 2 g cm$^{-3}$ was used to convert the aerodynamic diameter

measured by the APS to mobility diameter prior to merging the two size distributions. A condensation particle counter (CPC; TSI model 3010; ARM Climate Research Facility, 2007, 2011), which has a ~10% detection efficiency for particles of 7 nm diameter (Mertes et al., 1995), was connected to the same inlet as the SMPS+APS. The CPC data were used to



augment the size distribution data at the smallest particle sizes, as described in the Appendix, to result in number

concentrations for $D_p$ ranging from 7 nm to ~14 μm. The details of the ARM data streams used, the multiple quality control

tests performed, the size distribution adjustments made that incorporated the CPC data, and a validation of these adjustments

are also included in the Appendix. Of the 5 years of archive data that were processed, over 3 years of data (15,202 2-hour

samples) passed our quality control process and were used in the subsequent analyses. The resulting dataset that was utilized

in this study is shown in Figure 1. Gaps in the data timeline represent time periods with unavailable data or data that did not

pass quality control tests. The largest gap in the data (October 2010 through April 2011) was due to an internal leak in the

CPC that was documented in the ARM dataset. While the SMPS+APS data were available during this period, the CPC

adjustments could not be made and therefore, these data were excluded from this study.

## 10    3 Seasonal Variations in Aerosol Concentrations

Several previous studies have found strong seasonal differences in aerosol properties at the SGP site (e.g., Andrews

et al., 2011; Parworth et al., 2015; Sherman et al., 2015), and we therefore used the same season definitions (MAM, JJA,

SON, DJF) as these prior studies in order to facilitate comparisons. The 25[th], 50[th], and 75[th] percentile aerosol number size

distributions were computed for each season as well as for the entire 5-year period (ALL) and are shown in Figure 2a; these

number distributions were converted to surface area (S) and volume (V) size distributions as shown in Figures 2b and 2c.

While similarities are evident in the seasonal size distributions' shapes and modes, several differences between the seasons

can be seen in Figure 2. JJA had a higher fractional contribution of particles with diameters larger than 50 nm as compared

to the other seasons, which led to higher total surface area and volume concentrations in JJA. MAM and SON more

frequently had larger concentrations of the smallest particles ($D_p < 20$nm), while DJF often had very few small particles. Four

lognormal distribution modes were found to best fit the median size distributions (Figure 3), where the lognormal

distribution was defined as:

$$N\left(\ln\left(D_p\right)\right) = \int_{\ln(D_p)}^{\ln(D_p)+d\ln(D_p)} \frac{N_0}{\ln(\sigma_g)\sqrt{2\pi}} e^{-\frac{(\ln(D_p)-\ln(D_m))^2}{2\ln^2(\sigma_g)}} \, d\ln D_p \qquad (1)$$

where $N(ln(D_p))$ is the number concentration of aerosol particles between $\ln(D_p)$ and $\ln(D_p)+d\ln(D_p)$, $N_0$ is a total number

concentration within the mode (# cm$^{-3}$), $\sigma_g$ is the geometric standard deviation, and $D_m$ is the median diameter (μm). One

lognormal mode, as opposed to two, was chosen to fit the coarse mode because the decrease in concentrations around 3 μm

was a data artifact. The fitting was completed such that the mode parameters (Table 1) were converted between the number,

surface area, and volume size distributions, and the integrated number and surface area were within 1% of the observed

median values. The integrated volume values from the fitted distributions were ~2-4% higher than the median distributions

values due to the aforementioned data artifact. The parameters for the number size distributions are shown in Table 1. The

persistent but highly variable presence of a sub-30 nm mode, not completely resolved by the instrumentation at SGP, was

likely associated with the growth of newly formed aerosol particles into the size ranges that were observed by the instrument



suite used here. The next two modes approximate Aitken and accumulation modes with lognormal number distribution median diameters of 50-65 nm and 150-175 nm, respectively. Finally, one coarse mode represents the supermicron aerosol particles. It is important to note that the location and steepness of the drop-off in the largest aerosol mode may be related to the upper limit of the APS, as well as the decrease in inlet transmission efficiency for the largest particles. The resulting 4

regions of the aerosol size distribution are demarcated by the vertical grey lines in Figures 2 and 3 and represent particles with $D_p$ between 7 and 30 nm, 30 and 140 nm, 140 and 800 nm, and 800 nm and 14 μm. The integrated number concentrations within these 4 size ranges ($N_{7-30nm}$, $N_{30-140nm}$, $N_{140-800nm}$, and $N_{800nm+}$) are used for further analyses in this study. While the focus of this study is primarily on number concentrations, we have performed the same analyses for the same aerosol modes for integrated surface area and volume concentrations. Generally, the results were consistent amongst

the integrated number, surface area, and volume distributions. These analyses are included in the supplement for completeness.

To better quantify the variability within a season as well as the differences between seasons, Figure 4 shows the distributions of total measured aerosol number concentrations of particles between 7 nm and 14 μm ($N_T$) for the entire period (ALL) and for each season, as well as the integrated number concentrations for each of the 4 size ranges. To estimate the

statistical significance of the differences between the seasonal distributions, a simple bootstrapping technique was used. For each season, the effective sample size was estimated using lag-1 autocorrelations (Leith, 1973; Wilks, 2011) since the 2-hour samples were not independent. This typically reduced the sample size by a factor of 0.04-0.29, depending on the lag-1 autocorrelation of each integrated variable in each season. 10,000 random samples of a size equal to the effective sample size for each season were drawn, with replacement, from the ALL distribution. For each of the 10,000 random samples, the

mean, median, interquartile range (IQR), and the 5% and 95% percentile range (R595) were calculated, resulting in a distribution of these summary statistics for the 10,000 random samples. Then, the mean, median, IQR and R595 were computed for each season's data and were compared to the distribution of random samples from ALL. For example, the percentile of the DJF mean concentration for $N_T$ was 1% (grey diamond in the top row of Fig. 4f). In other words, when 10,000 random samples of the ALL $N_T$ data were taken with the effective sample size of the DJF $N_T$ data, only 1% of those

10,000 samples had means smaller than the DJF mean, suggesting the DJF mean value is significantly different from (in this case significantly less than) the ALL mean value. Bolded distribution characteristics in Figure 4a-e represent instances where that key statistic was less than the 5[th] percentile or greater than the 95[th] percentile of the distribution of random samples from the ALL data, suggesting significantly lower and higher values than the ALL data, respectively. It is important to note that these are arbitrary levels of significance, and Figure 4f shows the entire range of percentile values for each

distribution statistic for all the integrated number variables. We have also included the same analysis for surface area and volume distributions in the Supplement.

In terms of total aerosol number concentrations ($N_T$, Figure 4a), the DJF mean (5195 cm$^{-3}$) and median (3808 cm$^{-3}$) concentrations were significantly lower than ALL, while the median SON value (4572 cm$^{-3}$) was significantly higher than the other time periods. MAM was the most variable season, with a significantly different IQR and R595, while JJA was





significantly less variable than the other time periods, with a lower IQR and R595. For example, the R595s were 14286 cm$^{-3}$, 16889 cm$^{-3}$, 11957 cm$^{-3}$, 14072 cm$^{-3}$, and 13772 cm$^{-3}$ for ALL, MAM, JJA, SON, and DJF, respectively. These R595 results are consistent with the results of Sheridan et al. (2001), particularly their Figure 5a, which showed the largest breadth of number concentrations in the spring months and smallest breadth in the summer months. These results suggest the

importance of seasonal synoptic scale weather variability with respect to $N_T$ variability. For example, Andrews et al. (2011) used back trajectories to determine the transport pathways of aerosol to the SGP site, and in the MAM, SON and DJF periods, there were high frequencies of pathways coming both from the northwest and from the south or southeast, while in JJA the pathways were primarily from the same direction (southerly), resulting in lower variability in observed aerosol properties. Furthermore, several studies have documented episodically high concentrations of aerosol particles at SGP in

MAM from both local agricultural / wildfire sources and from the transport of biomass burning aerosol into this region from various parts of North America (e.g., Peppler et al., 2000; Wang et al., 2009).

    For $N_{7-30nm}$, the MAM mean value (3512 cm$^{-3}$) was the largest of all seasons, while the SON median value (1669 cm$^{-3}$) was the largest, demonstrating the MAM had the most extreme high concentrations of particles within this smallest size mode, while high concentrations were more frequent during SON. JJA had a significantly lower mean (2639 cm$^{-3}$) value

for total concentrations within this mode, as well as significantly lower variability in terms of lower IQR (2196 cm$^{-3}$) and R595 (10315 cm$^{-3}$), as compared to the other time periods, which may have been a result of a consistent coagulation sink due to the higher concentrations of larger aerosol (Figure 2). DJF had the highest frequency of low concentrations, which lowered the median concentration (1080 cm$^{-3}$). This smallest size mode was also associated with the highest variability of all the aerosol modes (in terms of absolute values) as seen by the breadth of the R595 (spanning several orders of magnitude).

This large variability was likely caused by the frequent bursts of high concentrations associated with new particle formation and the growth of these newly formed particles into the size ranges observed in this study, although uncertainties associated with the observations of particles within this smallest mode may have also contributed to this variability, as discussed in the Appendix.

    For $N_{30-140nm}$, a shift in seasonal trends occurred. JJA, which had significantly lower concentrations than ALL for

$N_{7-30nm}$, had a significantly larger mean (2315 cm$^{-3}$) and median (2037 cm$^{-3}$) concentration, which could be related to enhanced precursor concentrations in the summer months (e.g., Parworth et al., 2015). A similar reversal of trends occurred for MAM, which had a significantly lower mean (1959 cm$^{-3}$) and median (1523 cm$^{-3}$) concentration for $N_{30-140nm}$ as compared to ALL. As was the case for $N_{7-30nm}$, JJA was the least variable season for $N_{30-140nm}$. The seasonal trends for $N_{140-800nm}$ were similar to $N_{30-140nm}$, albeit with smaller differences between the seasons.

There was large seasonal variability associated with concentrations of the largest particles ($N_{800nm+}$). JJA had a significantly higher mean (1.53 cm$^{-3}$) and median (0.85 cm$^{-3}$) concentration and had significantly higher variability (R595 of 5.32 cm$^{-3}$), as compared to the other seasons. On the other hand, SON had a significantly lower mean (0.69 cm$^{-3}$) and median (0.44 cm$^{-3}$) concentration and significantly lower variability (R595 of 1.79 cm$^{-3}$), as compared to ALL. MAM also had significantly lower variability (R595 of 2.07 cm$^{-3}$). Interestingly, while DJF had a significantly low median concentration



(0.50 cm$^{-3}$) as compared to ALL, its mean concentration (1.27 cm$^{-3}$) was larger than the ALL data mean (1.06 cm$^{-3}$), due to the presence of a few time periods with very high concentrations within this mode. These $N_{800nm+}$ results are generally consistent with prior studies (Sheridan et al., 2001; Andrews et al., 2011), which have attributed the seasonal presence of coarse mode aerosol particles to dust, both from local sources and transported into the region.

## 4 Sub-Seasonal Cycles within Aerosol Number Concentrations

### 4.1 Methods

While the prior section was focused on seasonal differences in the aerosol size distribution, the focus of this section is the investigation of the sub-seasonal variability on time scales from several hours to several weeks using power spectral analysis. Power spectral analysis is a computational tool that fits a range of harmonic functions of varying frequencies to a data series using Fourier sums, and then calculates the amount of total variance in a data series that can be explained by each harmonic function, each associated with a specific frequency and period. The amount of variance explained by each frequency is often termed the power spectrum. The length and resolution of the data series on which the power spectral analysis is computed determines the frequencies of cycles within the dataset that can be resolved and tested. The cycle periods (T) and frequencies (f) that are resolved in such analyses are given by:

$$T = \frac{1}{f} = \frac{M}{k} \ , where \ k = 1, \dots , \frac{M}{2} \tag{2}$$

where $M$ is the length of the data series.

The aerosol number concentration data were separated into the 4 seasons as was done in Section 3. Then, the data were further partitioned into years to ensure a continuous time series, a requirement for spectral analysis. This partitioning resulted in the following 21 data subsets JF-2009, MAM-2009, JJA-2009, SON-2009, DJF-2010, …, SON-2013, D-2014. The DJF seasons included the December month of the prior year to create the continuous time period. For each of these 21 subsets, anomalies were first recalculated as differences from the subset mean and the anomalous data were then separated into smaller data chunks (7 days and 28 days in this study) for spectral analysis. Two choices for the length of the data series ($M$) were used in order to study different temporal scales. The resulting power spectra were averaged together by season for all the years and tested for significance. Separating each of the 21 seasonal subsets into smaller data chunks and averaging the resulting power spectra together increased the robustness of the analysis. Because of the difficulties in fitting harmonic functions at the edges of finite data, a Hanning window was applied to smooth the data. However, it should be noted that using such a smoothing method also limited the smallest frequency (largest period) that could be accurately detected. In order to account for this smoothing and to incorporate all the data, a 50% overlap window was also applied to the data.

To determine the statistical significance of the averaged power spectra, red noise spectra were estimated from the data. For each length $M$ data chunk without any missing values, the lag-1 autocorrelation ($r_{lag1}$) was determined. The red noise power spectra were then computed for each data chunk using the following formula from Gilman et al. (1963):





$$red\ noise(f,r) = \frac{1-r_{lag1}^2}{1-2r_{lag1}cos(2\pi f)+r_{lag1}^2}^{\,2} \tag{3}$$

These red noise power spectra were averaged together for each season. The 99% confidence level was calculated using the F-distribution, with the test statistic being the ratio of variances (i.e., power) of the actual data to that of red noise at the same frequencies. The degrees of freedom used for calculating the 99% confidence level were based on the number of individual

power spectra that were averaged together multiplied by 2.8 (Welch, 1967) for the actual data spectra and 1000 for the red noise spectra. Choosing a relatively large value (1000) for the red noise degrees of freedom demonstrates confidence in our red noise spectrum formulation. However, other values (100, 500) were tested and resulted in no qualitative changes to the results presented herein.

### 4.2 Hourly-to-Daily Cycles of Aerosol Number Concentrations

To determine the hourly-to-daily power spectra, the data series were binned and averaged over 2-hour intervals, with a length of the data series ($M$) of 7 days, thus resolving 4-hour to 3.5-day cycles in the data. Missing data for up to 6 hours were interpolated linearly from surrounding values. The resulting power spectra for total aerosol concentrations ($N_T$), for the entire period and by season, are shown in Figure 5. The strongest cycle in this aerosol dataset was the 24-hour or diurnal cycle. This was present in the average power spectrum for each season and for the entire dataset and always

exceeded the 99% significance level as compared to red noise. In other words, we can state with very high confidence that the diurnal cycle in these data did not arise from random fluctuations as represented by a red noise time series. 48%, 37%, 42%, and 42% of the total number of weekly data chunks had power associated with the diurnal cycle greater than that of red noise for MAM, JJA, SON, and DJF, respectively. Therefore, while MAM had slightly more frequent diurnal cycles in $N_T$, this diurnal cycle was a year-round phenomenon at the SGP site. All seasons, except JJA, also exhibited a 12-hour cycle in

$N_T$ at 99% confidence. We will first focus on the 24-hour cycle and then examine the 12-hour cycle in the following sections.

### 4.2.1. 24-hour (Diurnal) Cycle of Aerosol Particles

The subset of weekly data chunks that had power associated with the diurnal cycle greater than that of red noise was used to calculate the timing of the maximum and minimum aerosol concentrations associated with the diurnal cycle.

Although the focus here will be on the timing of the maximum concentrations, the timing of minimum concentrations can be calculated by shifting the maximum concentration timing by half of the period of interest (i.e., for the diurnal cycle, a 12-hour shift between maximum and minimum concentrations). Figure 6 shows the normalized frequency of the maximum aerosol concentrations associated with the diurnal cycle as a function of time. The maximum aerosol number concentrations associated with the diurnal cycle primarily occurred between 18 and 02 UTC (13 and 21 CDT). While the timing of the

diurnal cycle peak was generally in the local afternoon and evening hours for all seasons, the exact timing shifted between the seasons. The peak in the JJA diurnal cycle occurred several hours earlier (peak concentrations around 18-22 UTC or 13-





17 CDT) than the peak in the annual average (20-22 UTC or 15-17 CDT), and the peak for DJF was shifted towards the later hours (peak concentrations from 20-02 UTC or 15-21 CDT) relative to the annual average.

To better understand the aerosol processes related to this diurnal cycle in $N_T$ and to test whether there were size-dependent cycles, power spectra for the integrated aerosol number concentrations for each of the 4 modes of the aerosol size distribution ($N_{7-30nm}$, $N_{30-140nm}$, $N_{140-800nm}$, and $N_{800nm+}$) were computed and are shown in Figure 7. There were statistically significant diurnal cycles for all seasons for $N_{7-30nm}$ and $N_{140-800nm}$. For $N_{30-140nm}$, JJA had the strongest diurnal cycle, although the diurnal cycles for $N_{30-140nm}$ were relatively weaker, as compared to red noise, than those for $N_{7-30nm}$ and $N_{140-800nm}$. For the largest particles ($N_{800nm+}$), there was no consistent diurnal cycle above that of red noise, although there was some enhanced power in JJA. These results were generally consistent for the integrated surface area and volume concentrations unless otherwise noted.

As was done for the total integrated number concentration for the entire size distribution, $N_T$, the timing of peak concentrations associated with the diurnal cycle was calculated for each of the 4 aerosol size ranges (Figure 8). Because small particles often accounted for the majority of the total number concentrations, $N_{7-30nm}$ was the primary driver of the diurnal signal in the total aerosol number concentrations ($N_T$, Figure 5). This was further corroborated by the fact that the timing of the diurnal cycle peak concentrations for $N_{7-30nm}$ occurred at approximately the same times as that for $N_T$ (compare Figure 8a with Figure 6). Aerosol particles in this smallest size range are typically presumed to have originated in new particle formation (NPF) events, followed by growth of those newly formed particles to sizes that can be detected by the instruments used in this study. Niemenen et al. (2018) assessed NPF at many sites around the world, including SGP, and found that the presence and growth of these small particles most frequently occurred in MAM (25% of the time) at SGP, but were much less frequent in the other seasons (10% in SON, 8% in DJF, and 4% in JJA). While our results corroborate the high concentrations of small particles in MAM, they also indicated consistent diurnal cycles of $N_{7-30nm}$ throughout the year. 55%, 46%, 56%, and 48% of the weekly $N_{7-30nm}$ data chunks had 24-hour cycles with power above that of red noise for MAM, JJA, SON, and DJF, respectively. Reasons for differences between this study and Niemenen et al. (2018) are likely related to the incorporation in this study of CPC data and the adjustments made to the aerosol size distribution at these smaller sizes (see Appendix), but are also related to the metric used to assess the presence of these small particles.

The broadly consistent timing of the diurnal cycle in $N_{7-30nm}$ throughout the year (local afternoon/evening) may suggest similar formation, growth, and/or transport mechanisms for aerosol with $D_p$ between 7 and 30 nm. The several-hour seasonal shift in the timing of the peak concentrations between seasons may also help elucidate some of the processes leading to observations of elevated $N_{7-30nm}$ at the SGP surface site. At SGP, the height of the atmospheric boundary layer reaches a specified altitude earlier in JJA and later in DJF, with MAM and SON falling in between (Liu and Liang, 2010; Delle Monache et al., 2004). Therefore, the shift in timing of the diurnal cycle of $N_{7-30nm}$ found in this present study corroborates earlier work that suggested nucleation of new particles sometimes occurs in the free troposphere or residual layer and is observed at the surface when mixing processes transport these aerosol to the surface (e.g., Weingartner et al., 1999; Hallar et al., 2011; Chen et al., 2018). This shift in timing may also be related to the seasonal shifts in insolation,

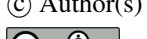



including both the variation in sunrise times and intensity, and the resulting impacts on photochemical processes leading to the formation and growth of small aerosol particles (e.g., O'Dowd et al., 1999).

For $N_{30-140nm}$, there was a weaker diurnal signal in all seasons (Figure 7f-j). The timing of the peak concentrations often occurred in the night and early morning hours, several hours after the peak in concentrations of $N_{7-30nm}$. This signal

5 could be representative of the growth of the $N_{7-30nm}$ aerosol mode to larger sizes. It is important to note that timing of peak concentrations of the diurnal cycle associated with these particles was more variable (Figure 8b) than for $N_{7-30nm}$, with peak concentrations occurring at almost all times of the day. Therefore, the timing of and processes associated with the diurnal cycle for $N_{30-140nm}$ were much less consistent throughout this dataset and could be related to a wide range of aerosol, radiative, and dynamical processes.

10 For $N_{140-800nm}$, a more consistent diurnal cycle was present for all seasons (Figure 7k-o). The timing of the $N_{140-800nm}$ diurnal cycle was also generally consistent for all the seasons, with peak concentrations occurring between 08 and 16 UTC (03 and 11 CDT). These results are consistent with those for the integrated volume concentration for this mode ($V_{140-800nm}$, Figs. 9k-o and 10c), with volume concentrations providing a better comparison to prior studies that focused on optical properties and aerosol mass concentrations. For example, the timing of the diurnal cycle in $N_{140-800nm}$ (and $V_{140-800nm}$) was

15 similar to the reported diurnal cycle in the light absorption coefficient for $D_p < 10$ μm (Sheridan et al., 2001) and nitrate and organic aerosol mass concentrations for submicron particles from December 2011 through May 2011 (Parworth et al., 2015). To explain this diurnal cycle in particles between 140 and 800 nm, data from an Aerosol Chemical Speciation Monitor (ACSM) at the SGP site (Ng et al., 2011) from August 2011 through December 2013 was used. The data was filtered to only include weekly data with power associated with the $V_{140-800nm}$ diurnal cycle that was greater than that of red noise. The

20 ACSM measured non-refractory submicron aerosol mass concentrations for several species, including nitrate, sulfate, ammonium, and organic aerosol. The timing of peak ACSM total mass concentrations (Figure 11) aligns with the timing of peak concentrations in $V_{140-800nm}$ and $N_{140-800nm}$ (Figure 10c and 8c, respectively). The ACSM data demonstrate that the diurnal cycle in $V_{140-800nm}$ was related to nitrate and organic aerosol mass concentrations, although their relative contributions to the diurnal cycle varied by season. Organic aerosol had much stronger diurnal variations in JJA as compared to nitrate,

25 while nitrate had stronger diurnal variations in DJF. Ammonium also had a similarly timed cycle in MAM, SON, and DJF, but with much lower anomalous concentrations. These trends represent a variety of aerosol processes, including temperature-dependent gas-to-particle partitioning, regional aerosol transport, and local emissions, and generally agree with the results of Parworth et al. (2015). Focused modelling studies and measurements are needed to further determine the specific and most important pathways leading to these diurnal cycles in aerosol concentrations.

30 Lastly, while there were no significant diurnal cycles in $N_{800nm+}$ (Fig. 7p-t), there were significant peaks for the diurnal cycle associated with the integrated volume of particles within this size range ($V_{800nm+}$, Fig. 9p-t), with the strongest signals in MAM and DJF. The timing of peak concentrations associated with the diurnal cycle in $V_{800nm+}$ was consistent amongst seasons and primarily occurred during the local evening hours, between 22-24 UTC (17-19 CDT, Figure 10d). The fact that this signal was weaker in $N_{800nm+}$ suggests that the diurnal signal was primarily associated with the largest particles



within the coarse aerosol mode. This result aligns with the results of Andrews et al. (2011), which documented low Ångström exponent values in their spring and winter measurements at SGP, which is often a signal for large dust aerosol. Also, surface meteorology data from the SGP site (ARM Climate Research Facility, 1995) during the same 5-year period demonstrate that surface winds, on average, reach a peak between 20 and 24 UTC, with stronger winds occurring in MAM

and DJF. Therefore, we speculate that the timing of the $V_{800nm+}$ diurnal cycle was related to the timing of strong wind conditions, which can loft large aerosol particles.

### 4.2.2. 12-Hour Cycle of Aerosol Particles

The strongest cycle with respect to red noise in the $N_T$ data was the diurnal cycle (Figure 5). However, there was also a statistically significant 12-hour cycle present in some of these data, particularly in MAM and DJF (Figure 5b,e). In

general, the variability in $N_T$ was caused by variability in $N_{7-30nm}$, due to the high concentrations and high variability of particles in this size range. The peak concentrations of the 12-hour cycle for all seasons occurred between 04 and 12 UTC (23 and 07 CDT) and between 16 and 24 UTC (11 and 19 CDT) for both $N_T$ and $N_{7-30nm}$. The similarities between the timing of the peak concentrations of the 12-hour cycles for $N_T$ and $N_{7-30nm}$ further demonstrate the regulating relationship that $N_{7-30nm}$ has on $N_T$.

The latter of the two daily peaks in concentrations associated with the 12-hour cycle occurred at approximately the same time as the peak concentrations associated with 24-hour cycle (16-02 UTC or 11-21 CDT), suggesting that the 12- and 24-hour cycles are related. To explain this relationship between the 12- and 24-hour cycles, Figure 12 shows the weekly aerosol data (22-29 February 2012) that had the strongest 12-hour cycle, broken down into their 12- and 24-hour cycle components. The peak concentrations of the 24-hour cycle (yellow) clearly aligned with the peak concentrations of the

aerosol data (black). However, the minimum in aerosol concentrations typically occurred directly before peak $N_{7-30nm}$, as opposed to the 12-hour shift that would be associated with a purely diurnal cycle. When including the 12-hour cycle (cyan), the combination of the 12- and 24-hour cycles (green) much better represented the aerosol time series (black). Therefore, the power associated with the 12-hour cycle manifested from the different rates of growth and decay of aerosol number concentrations. The formation of $N_{7-30nm}$ occurred at a much faster rate than the loss of $N_{7-30nm}$. While the 12-hour cycle

primarily manifested from the sudden increase in number concentrations in this size range, it is important to note there were also time periods where a second peak in $N_{7-30nm}$ occurred in the 04-12 UTC (23-07 CDT) time frame (e.g., 26-27 Feb 2012 in Figure 12).

### 4.3 Daily-to-Weekly Aerosol Cycles

Several prior studies have demonstrated weekly cycles in aerosol total number concentrations (Sheridan et al, 2001)

and aerosol optical properties (Delene and Ogren, 2002; Sherman et al., 2015) at the SGP site. Spectral analyses aimed at resolving cycles on the order of 2 days to 14 days required re-partitioning of the data into daily samples and 28-day data chunks. In order to achieve a larger number of 28-day continuous samples, the dataset was doubled to include the time



period between 1 January 2007 and 1 January 2017. However, since the SMPS+APS size distribution data were not available during this extended time range, only the total aerosol number concentrations from the CPC were used. The CPC data for this extended time range were screened in the same manner as was done for the earlier analyses and as described in the Appendix. Figure 13 shows the power spectra for the entire period and by season for the expanded dataset. For the entire

dataset, no cycles significant at the 99% confidence interval were found. However, the power spectra for MAM and SON had peaks just below this significance level for 7-day cycles, and the SON and DJF power spectra had peaks just missing this criterion for cycles lasting ~3.5-5 days. In JJA, there was no clear peak in the power spectrum above that of red noise on the time scales of 2-14 days. These results are possibly related to the temporal cycles of synoptic conditions and air masses in the southern United States. At the SGP site, JJA is typically associated with large-scale ridges and weak synoptic flows

(Coleman and Rogers, 2007) that would lead to stagnant air masses and no consistent cycles on these time scales. Using four years of springtime data, Lanicci and Warner (1991) determined that changing synoptic patterns lead to an approximately one week cycle in elevated mixed layers in the southern United States, and therefore, this periodicity in synoptic patterns could help explain the weak weekly cycle in MAM. These results are also consistent with the higher intraseasonal variability observed in MAM, SON, and DJF for $N_T$ (Figure 2). Other studies have corroborated our hypothesis about the importance of

synoptic scale variability on aerosol concentrations at SGP. For example, Power et al. (2006) demonstrated significant differences in aerosol optical depth based on the classified air mass present at many locations across the United States, including at SGP.

**5 Conclusions**

         The focus of this study is on 5-year (2009-2013) measurements from several instruments located at the Department
of Energy's Atmospheric Radiation Measurement's Southern Great Plains (SGP) site. These instrument datasets were merged to provide aerosol number size distributions for particles with diameters between 7 nm through ~14 μm and were also converted to surface area and volume size distributions. This quality-controlled dataset was used for two purposes. First, we provided key characteristics of the size distributions, including fits for 4 lognormal modes, both for the entire period and on a seasonal basis for the SGP site (a North American, rural, continental site). These observational data and analyses may

be useful for validating models that explicitly represent aerosol processes. Furthermore, the characteristic aerosol size distributions presented in this study could also be used in a variety of applications, including more realistic representations of aerosol activation, radiation, and ice nucleation, especially in models that do not have detailed aerosol processes. Second, we quantified the variability in aerosol concentrations, with a focus on number concentrations, for a range of time scales from hourly to seasonal. Variability in the total number concentrations, as well as the integrated concentrations within

specified size ranges that were associated with the different aerosol modes, was assessed.

         In terms of seasonal differences, for total aerosol number concentrations ($N_T$), MAM and SON had the largest mean concentrations, and DJF had the lowest mean concentrations. JJA had the lowest variability in $N_T$, as compared to the other





seasons, suggesting more consistent background aerosol conditions during the summer months. Comparing the integrated number concentrations within the aerosol modes, the variability in total number concentrations ($N_T$) was driven by the large variability in the smallest particles ($N_{7-30nm}$), which was likely related both to the presence of new particle formation events and the growth of these particles. JJA had the lowest mean concentrations of smallest particles ($N_{7-30nm}$), possibly due to a

coagulation sink that was associated with the fact that JJA had the highest mean concentrations of larger particles ($N_{30-140nm}$, $N_{140-800nm}$, and $N_{800nm+}$). The distributions of $N_{7-30nm}$ and $N_{800nm+}$ were more different between the seasons, as compared to $N_{30-140nm}$ and $N_{140-800nm}$. Therefore, the formation mechanisms and/or transport pathways of the smallest and largest particles have significant seasonal dependencies.

We used power spectral analyses to determine the presence of key temporal cycles within the aerosol size

distribution data. A predominant 24-hour (diurnal) cycle in each season was observed for $N_T$, driven by concentrations of the smallest particles ($N_{7-30nm}$). Peak concentrations associated with this diurnal cycle in $N_{7-30nm}$ and $N_T$ generally occurred in the afternoon and evening hours, with a slight seasonal shift in the timing that was consistent with seasonal shifts in insolation and boundary layer development. There was also a consistent diurnal cycle in $N_{140-800nm}$ (and $V_{140-800nm}$), with peak concentrations typically occurring between 08 and 16 UTC (03 and 11 CDT) in all seasons, consistent with the prior studies

that have focused on aerosol optical properties and mass concentrations and likely related to nitrate and organic aerosol mass concentrations. Because size-resolved measurements for a longer time period were unavailable, cycles in aerosol number concentrations for periods of days to weeks were tested only for $N_T$. Although there was no cycle that was sufficiently consistent to pass our 99% significance testing, there were several temporal scales that exhibited enhanced power, which varied by season and were likely related to synoptic scale variability at SGP.

While this study provided key characteristics of aerosol size distributions at SGP and quantified the temporal variability of aerosol number concentrations within varying sizes and on a range of scales (hourly-to-seasonal), there are still uncertainties in attributing this variability to physical mechanisms, for which more in-depth analyses are required. For example, the recent New Particle Formation Study (NPFS) (Smith and McMurray, 2015; NPFS, 2017), which took place in April-May 2013 at the SGP site, was focused on understanding the pathways under which aerosol particles are formed and

grow to larger sizes. Using the NPFS data, Hodshire et al. (2018) and Chen et al. (2018) presented several different growth pathways of newly formed particles during the 2013 spring period. Our study demonstrates with 5 years of observations that new particle formation and growth at SGP also occur frequently throughout the year, and therefore, new particle formation and the subsequent growth pathways at SGP may be a more significant contribution to cloud condensation nuclei than previously appreciated. Classifying specific time periods when there are both consistent cycles in the data and hypotheses as

to the mechanisms involved, as has been done in this study, can provide the temporal map for further detailed analyses using the wide range of instruments present at the SGP site or in future field campaigns.

*Data availability*. All data is publically available via the U.S. Department of Energy's Atmospheric Radiation Measurement (ARM) user facility data archive (https://www.arm.gov/data).



## Appendix A: Merged Aerosol Size Distributions

Five years (2009-2013) of data from three aerosol instruments at the ARM-SGP site were merged in order to create the aerosol size distribution dataset used in this study. One dataset was the aerosol size distribution data from the scanning mobility particle sizer (SMPS), part of the TDMA system, which were combined with size distribution data from the

aerodynamic particle sizer (APS). The merged size distribution from those two instruments spanned the diameter (mobility) size range between ~12 nm and ~14 μm with 215 bins (Collins, 2010; ARM Climate Research Facility, 2010, 2015). The other dataset contained total aerosol number concentrations from a TSI 3010 condensation particle counter (CPC; ARM Climate Research Facility, 2007, 2011). Therefore, total aerosol number concentrations can be obtained from both the integrated SMPS+APS size distributions and the CPC measurements. Because there were very few particles larger than the

upper limit of the SMPS+APS measurements and the CPC measured smaller particles than the SMPS+APS, concurrent CPC data were used to extend the SMPS+APS size distributions from ~12 nm down to 7 nm and to improve the representation of the aerosol size distribution at the smallest sizes, where the largest SMPS observation uncertainties exist. The details of the processing of these data are described here.

First, the CPC data were quality controlled. Data that were flagged by the ARM quality control as suspect or

incorrect due to faulty instrumentation or operation were removed. Also, CPC data that were consistently lower than the concentrations from a collocated cloud condensation nuclei counter (single column, DMT Model 1) at the highest supersaturation available (typically ~1%) and CPC data with unrealistically small ($< 200$ cm$^{-3}$) or unrealistically large ($> 100,000$ cm$^{-3}$) aerosol number concentrations were removed. The quality-controlled CPC data were then time-interpolated to the midpoint time of each SMPS+APS measurement period (~45 min). Then, the SMPS+APS data were quality controlled

as follows, and suspect or incomplete SMPS+APS data were removed. Suspect or incomplete SMPS+APS data included instances when 1) the CPC data were unavailable or incorrect during a given SMPS+APS measurement period, 2) the integrated number concentration from the SMPS+APS was unrealistic, as noted above, 3) large portions of the SMPS+APS size distribution were missing, which occurred sporadically due to shifts in the instrument voltage, 4) there were unrealistic peaks in the size distribution, particularly at large particle sizes, and 5 there were peaks in integrated number concentrations

in the first measurement after the daily calibration, which were likely due to contamination from residual particles from the atomized calibration aerosol. These checks resulted in the removal of ~25% of the SMPS+APS distributions, with the majority of data removal due to not having simultaneous CPC and SMPS+APS measurements. Despite this reduction in data quantity, over 31,700 size distributions remained, which equate to ~3 years of data during the 2009-2013 time period.

In order to synthesize the quality-controlled CPC and SMPS+APS measurements into one merged dataset, five

steps were taken (Figure A1). First, the SMPS+APS size distributions were extrapolated from their smallest size bin (usually ~12 nm) down to 7 nm, the approximate smallest size for which the CPC observes a significant fraction of aerosol particles (~10%; Mertes et al., 1995). The five smallest available size bins in the SMPS+APS size distribution were fit with a polynomial of the functional form:





$$dN(D_p) = aD_p^2 + b, \qquad\qquad\qquad (A1)$$

where a and b are coefficients and $D_p$ is the particle size bin diameter in μm. The coefficients, a and b, were determined via least-squares regression for each SMPS+APS size distribution, and the resulting polynomial was used to extrapolate the size distribution down to 7 nm (Figure A1, Step 1). Several functional forms were tested for this extrapolation, and the form in

Eq. (A1) produced the best results. Since the CPC only detected a fraction of the particles less than 28 nm, we also applied the CPC detection efficiencies from Mertes et al. (1995) to scale down the extrapolated size distributions (Step 2 in Figure A1) in order to represent the size-resolved distribution that the CPC would observe. Therefore, the integrated number concentration from the resulting SMPS+APS size distribution represents an estimate of the same quantity reported by the CPC. The integrated number concentrations from the SMPS+APS size distributions after Step 2 were compared to the CPC

total number concentrations. Since these two instruments were generally unmonitored during their deployments, a number of unreported issues (e.g., clogging or a leak in the air flow) may have caused the derived concentration measurements from either one of the instruments to drift for some extended periods of time. Therefore, in Step 3, the 2-week rolling median percentage difference between the two instruments was calculated for the entire time series and used to correct for any systematic drifts between the two instruments. This 2-week rolling median calculation excluded times between 1800 and

2400 UTC, when we would potentially expect large differences between the instruments due to new particle formation events and growth. Because of the higher uncertainties associated with the SMPS+APS total integrated number concentrations, the SMPS+APS size distribution was always scaled up or down to the CPC concentrations. This scaling factor was typically within 50% (median value of 7.3% for the entire dataset), except for two periods (January-February 2009 and September-December 2013) when the median percentage differences were consistently greater than 50%.

20       After correcting for this systematic bias (Step 3), the remaining difference between the CPC and SMPS+APS total number concentrations was used to adjust the SMPS+APS number size distribution, such that the integrated number concentration from the SMPS+APS size distribution equaled the CPC value. This difference in the total number concentration was applied to the SMPS+APS size distribution using an exponential function, only for sizes below the diameter associated with the 95th percentile of the cumulative integrated number concentration (median value of ~200 nm),

and taking into account the CPC detection efficiencies (Figure A2). An exponential function was chosen because there were much larger uncertainties in the observed number concentrations and diameters of the smallest particles in the size distribution and therefore, the need to correct particle counts was most likely associated with errors in the data for the smallest particle sizes. These uncertainties were associated with the possible loss of small particles within the inlet, sampling lines, and/or instrument due to evaporation or deposition to walls, the extrapolation of the SMPS+APS size distribution,

uncertainties associated with the charging probabilities of the smaller particles in the SMPS+APS system, and small errors in the high voltage supplied in the SMPS, which can lead to substantial uncertainties in the sizing of the smallest particles observed. The aerosol size distribution above ~200 nm was not changed in this step. The final correction function (Figure A2, black line) was applied in an iterative manner, nudging the size distribution up or down in order to match the integrated number from the SMPS+APS size distributions to the CPC total number concentration (Step 4). The resulting aerosol size



distributions after Step 4 were scaled back up by the reciprocal of the CPC detection efficiencies (Step 5) to represent an estimate of the true aerosol particle size distribution and number concentration at each time.

       To validate the adjustment algorithm described above, the original and adjusted size distributions were compared to data from the New Particle Formation Study (NPFS) (Smith and McMurray, 2015; NPFS, 2017). NPFS took place at the

SGP site for ~6 weeks in April-May 2013, and during this study, measurements of aerosol particle size distributions were measured down to ~3 nm in the SGP Guest Facility, a few hundred meters away from the CPC and SMPS+APS measurements. We compared the integrated number concentrations for aerosol with diameters between 7 and 30 nm from the NPFS to the adjusted SMPS+APS size distributions during this period, since the majority of changes to the SMPS+APS size distributions occurred in this size range (e.g., Figure A2). By incorporating the CPC data via the steps described above, the

adjusted SMPS+APS distributions better captured the timing and magnitude of aerosol concentrations at these small particle sizes (Figure A3). The correlation coefficient for this comparison improved from 0.37 to 0.89 from the original data to the adjusted data. The SMPS+APS size distribution data above 30 nm remained relatively unchanged, since the majority of the adjustments were applied below 30 nm. This improvement of the SMPS+APS aerosol number size distribution data demonstrates the utility of having a suite of related aerosol instruments at the same site that can be compared and combined

to provide a more comprehensive representation of aerosol characteristics.

*Author contributions.* PJM performed the analyses presented. PJM, EJTL, and DC assisted with the data access. PJM, EJTL, DC, and SMK interpreted the raw aerosol data and developed the merged aerosol size distributions. PJM, EJTL, DC, SMK, and SCV assisted with the interpretation of the analyses, and PJM prepared the manuscript with contributions from all co-

authors.

*Competing interests*. The authors declare that they have no conflict of interest.

**Acknowledgements**

This work was supported in part by a National Science Foundation Graduate Research Fellowship Grant DGE-1321845 and

in part by the U.S. Department of Energy's Atmospheric System Research, an Office of Science, Office of Biological and Environmental Research program, under Grant No. DE-SC0016051. All data were obtained from the Atmospheric Radiation Measurement (ARM) Program sponsored by the U.S. Department of Energy, Office of Science, Office of Biological and Environmental Research, Climate and Environmental Sciences Division. We would also like to acknowledge Sam Atwood for his assistance in fitting the size distribution data with multiple lognormal distributions.

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





| | | ALL | MAM | JJA | SON | DJF |
|---|---|---|---|---|---|---|
| *Mode 1* | | | | | | |
| | $N_0$ | 2606 | 3083 | 2171 | 2910 | 1911 |
| | $D_m$ | 0.00530 | 0.00550 | 0.00550 | 0.00550 | 0.00450 |
| | $\sigma_g$ | 2.80 | 2.80 | 2.80 | 2.80 | 2.80 |
| | | | | | | |
| Mode 2 | | | | | | |
| | $N_0$ | 1883 | 1406 | 2049 | 1896 | 1929 |
| | $D_m$ | 0.05866 | 0.05426 | 0.06460 | 0.05459 | 0.05343 |
| | $\sigma_g$ | 1.82 | 1.81 | 1.76 | 1.78 | 1.84 |
| | | | | | | |
| Mode 3 | | | | | | |
| | $N_0$ | 352 | 395 | 452 | 391 | 362 |
| | $D_m$ | 0.16624 | 0.15416 | 0.16189 | 0.15605 | 0.17262 |
| | $\sigma_g$ | 1.53 | 1.56 | 1.56 | 1.54 | 1.54 |
| | | | | | | |
| Mode 4 | | | | | | |
| | $N_0$ | 0.791 | 1.244 | 1.100 | 0.576 | 0.486 |
| | $D_m$ | 0.82355 | 0.69573 | 0.85788 | 0.87508 | 0.88354 |
| | $\sigma_g$ | 1.97 | 1.99 | 1.93 | 2.00 | 1.94 |

**Table 1: Parameters for each mode of the fitted lognormal distributions for the number size distributions shown in Figure 3. $N_0$ represents the amplitude of the lognormal distribution and the total number concentration within the mode (# $cm^{-3}$), $D_m$ represents the median diameter (µm), and $\sigma_g$ represents the geometric standard deviation, all as denoted in Equation 1 in the text.**





**Figure 1: Time series of the final aerosol dataset used in this study following the quality control and the aerosol number size distribution adjustments, as described in the Appendix. Each row represents one year from 2009 through 2013. The shading represents the value of the number size distribution, dN dlnD$_p^{-1}$, as a function of diameter (left axis), and the black dots represent the total integrated number concentrations (N$_T$, right axis).**




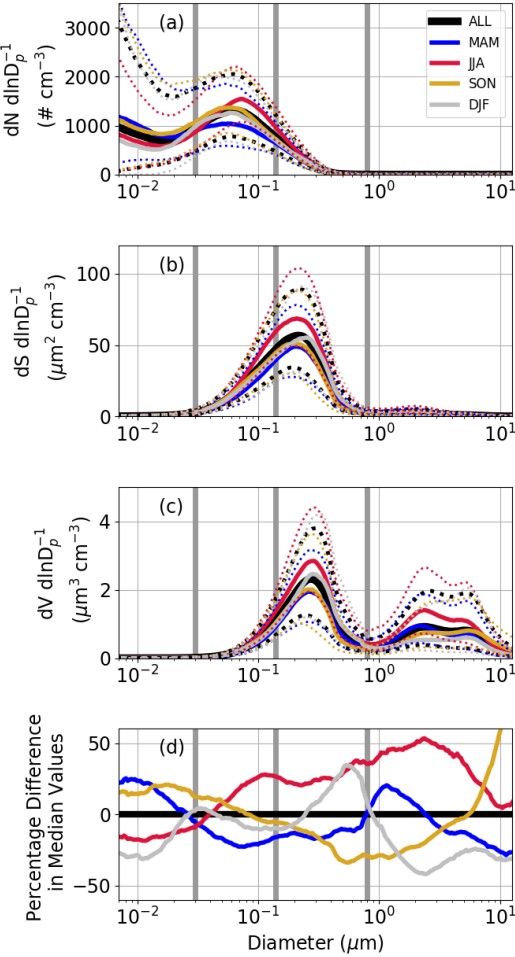

**Figure 2: Aerosol size distributions for the entire time period and by season. (a) represents the number size distributions (# cm⁻³), (b) represents the surface area size distributions (μm² cm⁻³), and (c) represents the volume size distributions (μm³ cm⁻³). The solid colored lines depict the median values, and the dotted lines depict the 25th and 75th percentiles. (d) represents the percentage difference in the median size distributions for each season with respect to the entire period (ALL). The vertical grey lines demarcate the four separate regions of the size distribution that were used for further analyses in this study.**



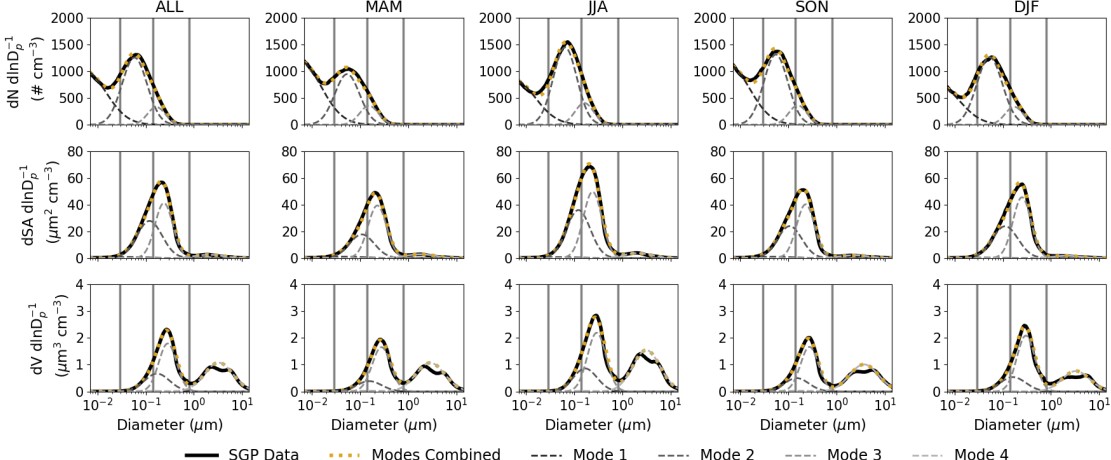

**Figure 3: Median distributions from each season (black) fitted with 4 lognormal distributions (modes). The columns (left to right) represent the time periods ALL, MAM, JJA, SON, and DJF, respectively. The rows (top to bottom) represent the number, surface area, and volume size distributions, respectively. The vertical grey lines demarcate the four separate regions of the size distribution that were used for further analyses in this study.**



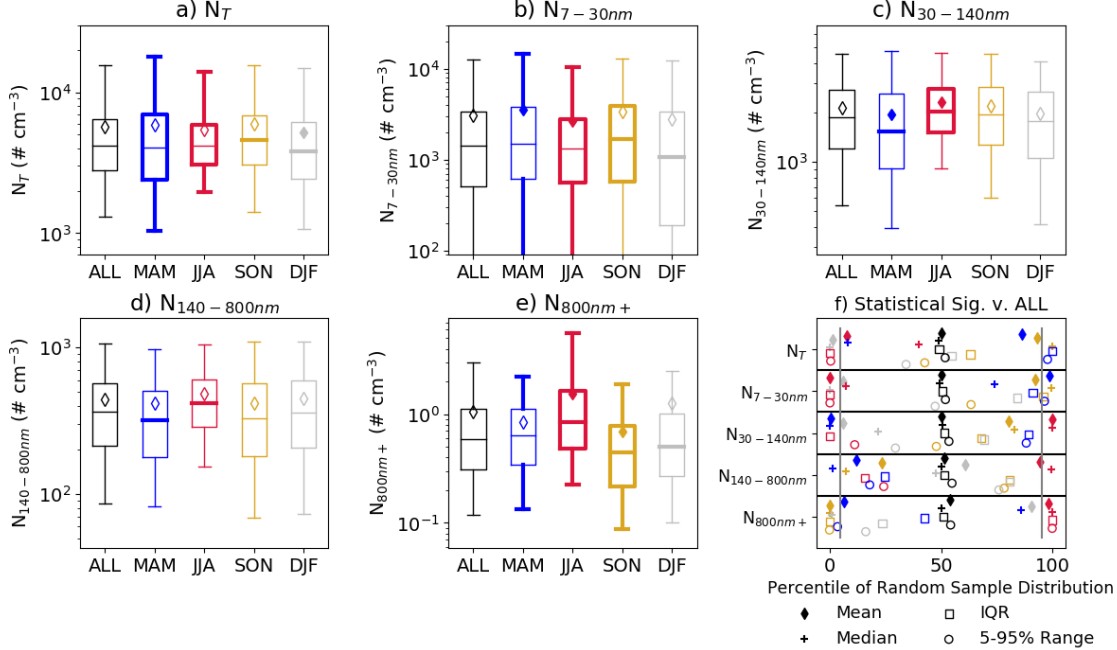

**Figure 4: Distributions of integrated number concentrations for the entire size distribution (a) and for the 4 size ranges (b-e, N$_{7-30nm}$, N$_{30-140nm}$, N$_{140-800nm}$, and N$_{800nm+}$), shown as box-plot diagrams. Data are shown for the entire time period (ALL) and by season. The boxes represent the interquartile ranges separated into two boxes by the median values, the diamonds represent the mean values, and the lines extending from the boxes represent the 5th and 95th percentiles. Bolded lines and solid symbols in panels (a) through (e) represent differences between the seasonal and ALL variables that are statistically significant at the 95% level, as described in the text and shown in panel (f). The vertical grey lines in (f) are the 5th and 95th percentiles.**


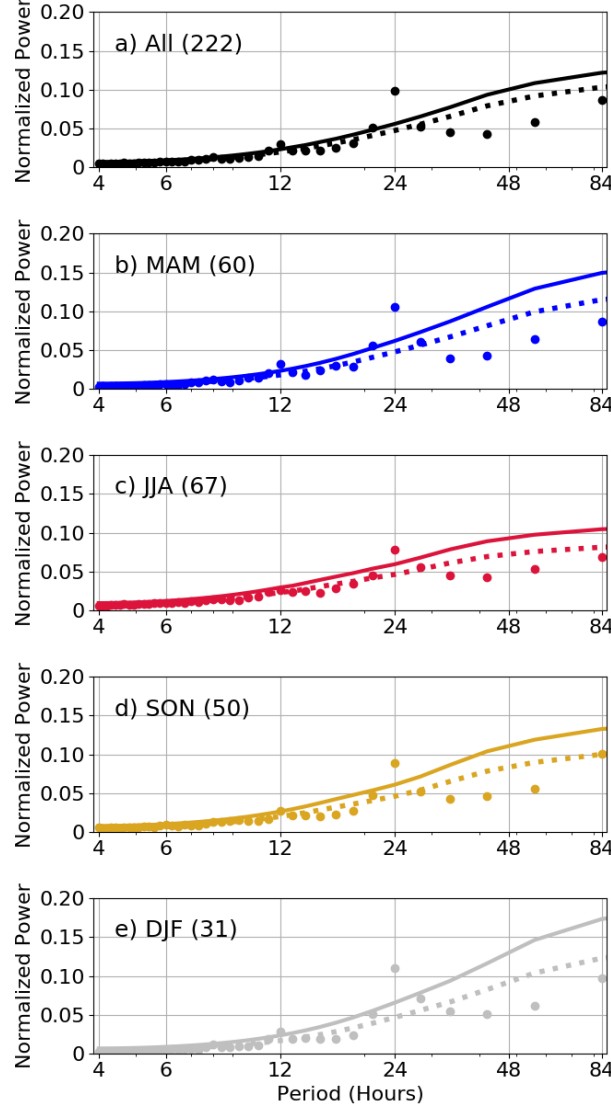

**Figure 5: Normalized power spectra for $N_T$ for the entire period (a) and by season (b-e). The dots represent power associated with the data. The dashed lines represent an estimate of the red noise power spectrum for each data set, and the solid lines represent the 99% significance testing level, as described in the text. The values in the parentheses are the number of weekly data chunks used in this analysis.**





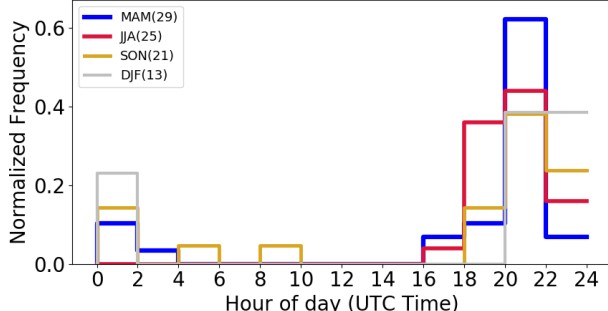

**Figure 6: Normalized frequency of the daily time of peak concentrations associated with the 24-hour cycle in $N_T$. This figure only includes weekly data chunks that had normalized power associated with the 24-hour cycle greater than that of the corresponding seasonal estimate of the red noise spectrum power. The numbers in parentheses represent the number of weekly data chunks that met this criterion.**





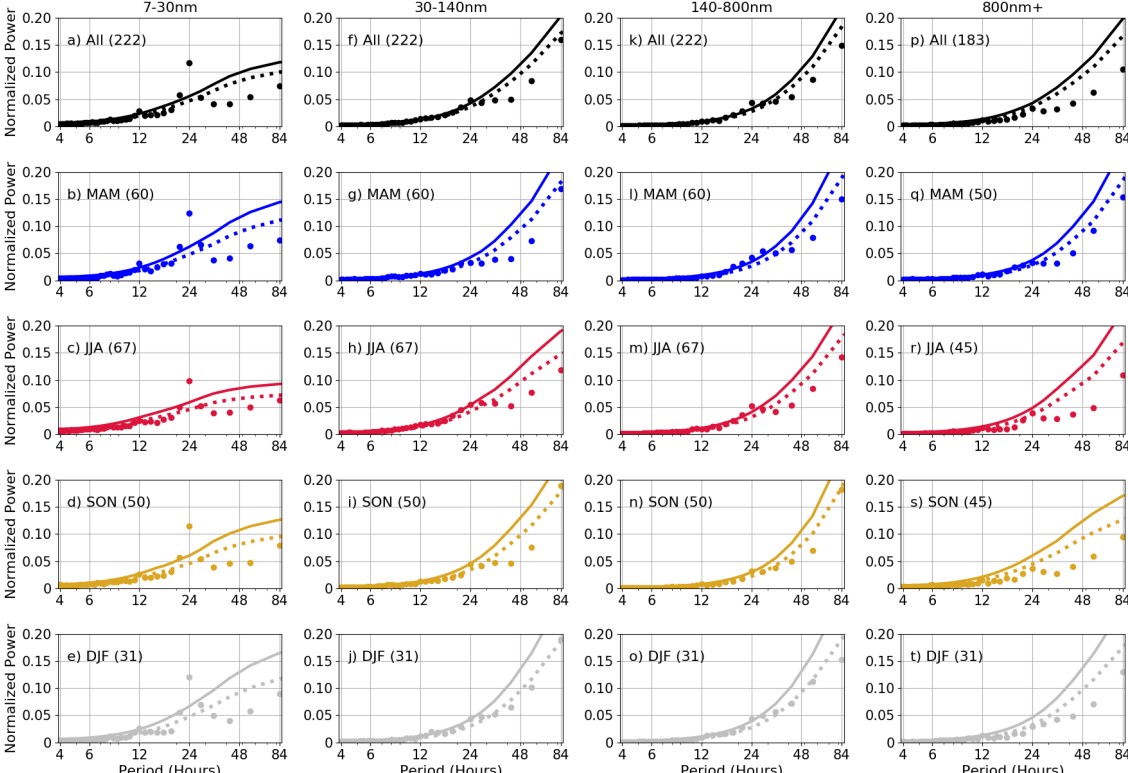

**Figure 7: Normalized power spectra for N₇₋₃₀nm, N₃₀₋₁₄₀nm, N₁₄₀₋₈₀₀nm, and N₈₀₀nm₊ for the entire period and by season. The descriptions of the symbols used are the same as in Figure 5.**


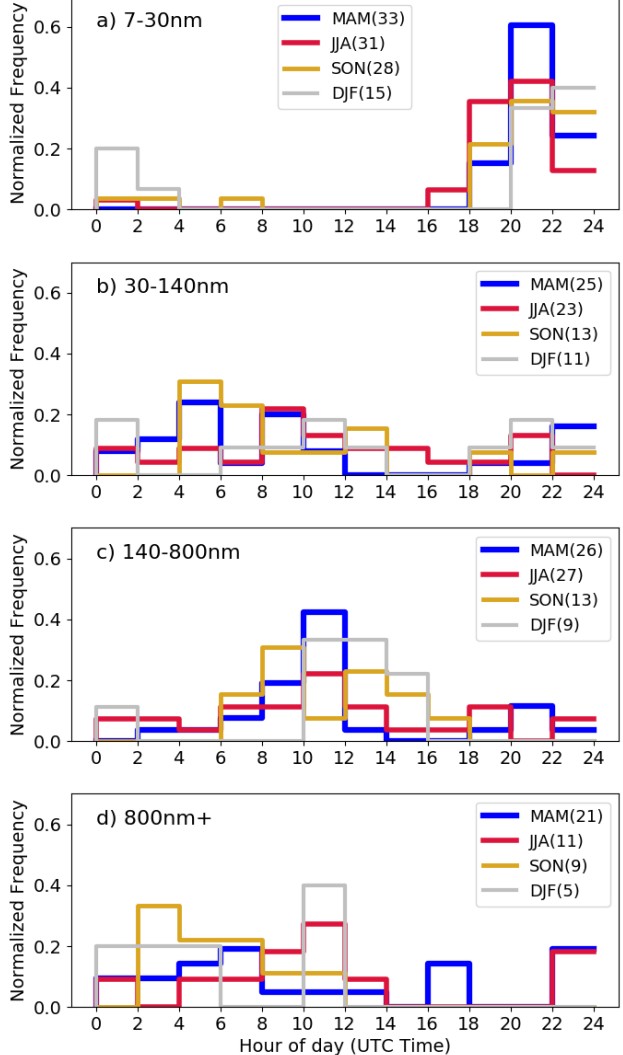

**Figure 8: Normalized frequency of the daily time of peak concentrations associated with the 24-hour cycle in the different modes of the aerosol number size distribution. (a-d) represent N$_{7-30nm}$, N$_{30-140nm}$, N$_{140-800nm}$, and N$_{800nm+}$, respectively. The description of the figure is the same as in Figure 6.**





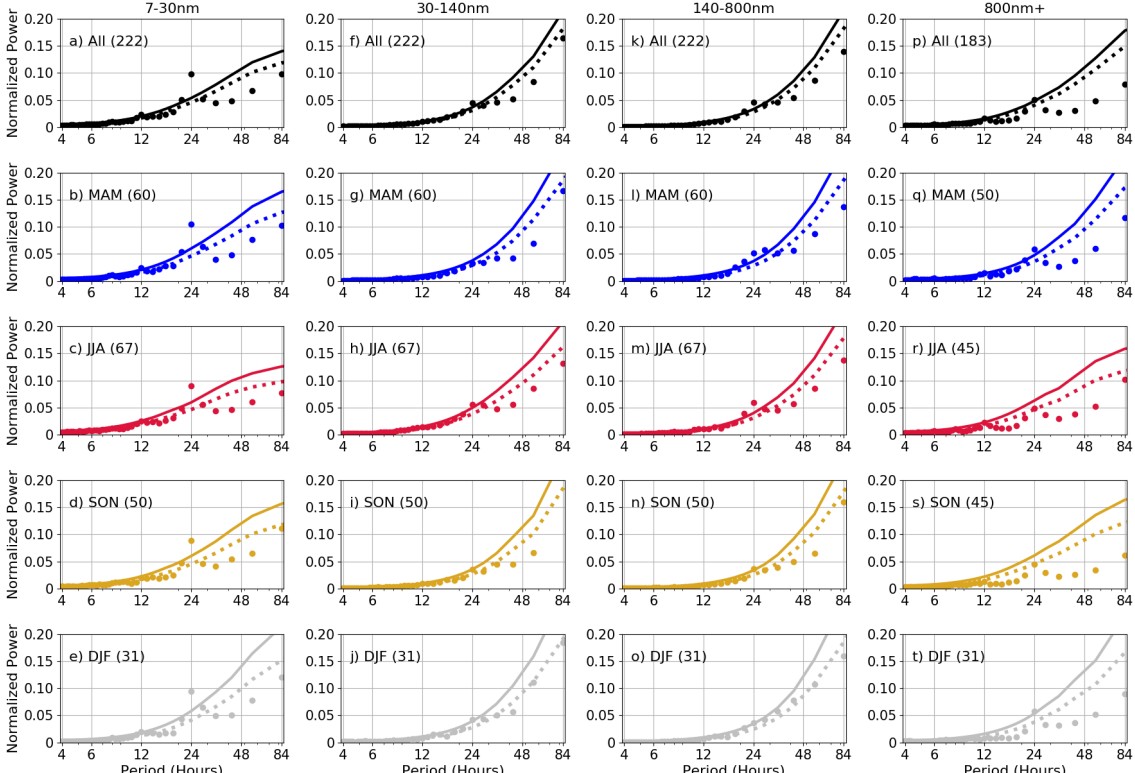

**Figure 9: Normalized power spectra for V$_{7\text{-}30nm}$, V$_{30\text{-}140nm}$, V$_{140\text{-}800nm}$, and V$_{800nm+}$ for the entire period and by season. The descriptions of the symbols used are the same as in Figure 5.**



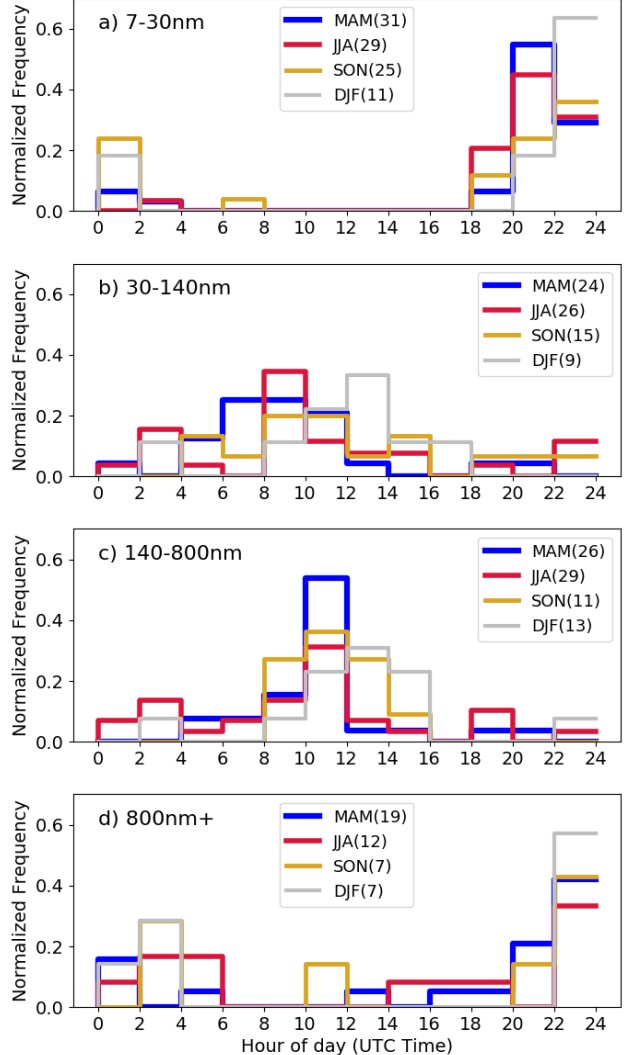

**Figure 10:** Normalized frequency of the daily time of peak concentrations associated with the 24-hour cycle in the different modes of the aerosol volume size distribution. (a-d) represent $V_{7\text{-}30nm}$, $V_{30\text{-}140nm}$, $V_{140\text{-}800nm}$, and $V_{800nm+}$, respectively. The description of the figure is the same as in Figure 6.



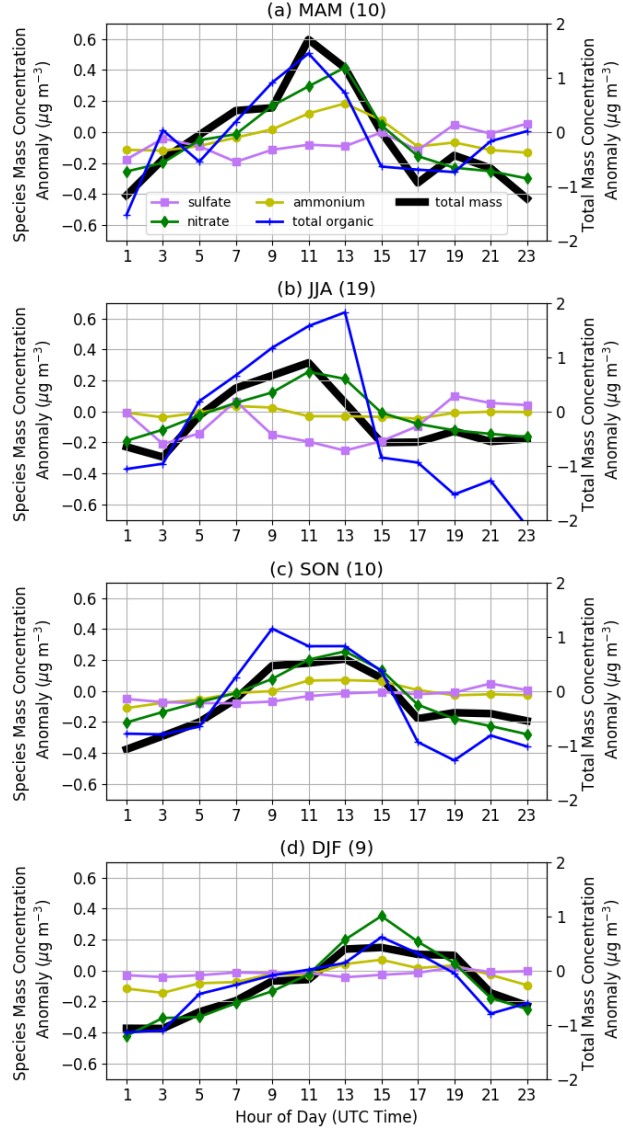

**Figure 11: Diurnal cycle of aerosol mass concentration anomalies for sulfate, nitrate, ammonium, and organic aerosol species (left axis) and total mass concentrations (right axis) from the ACSM. The data were separated into seasons (a-d) and only included the weekly time periods where the power associated with the 24-hour cycle in integrated volume between 140 and 800nm ($V_{140-800nm}$) was greater than that of red noise. The number of these weekly time periods is shown in the parenthesis in the panel titles.**





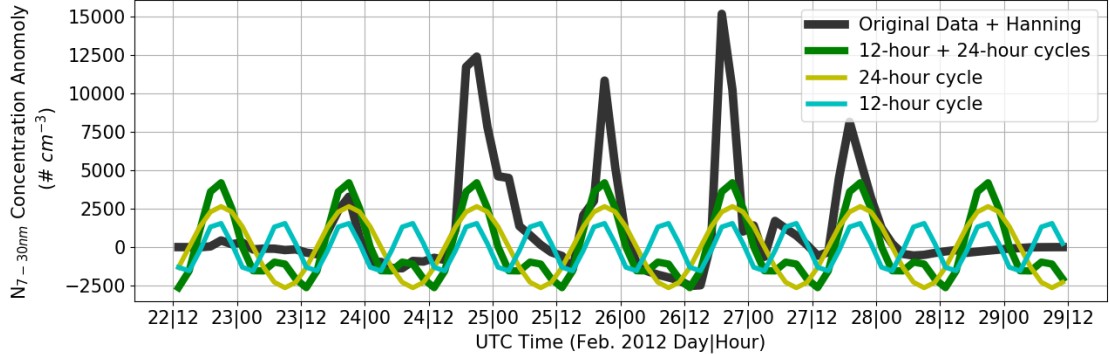

**Figure 12: N$_{7\text{-}30nm}$ for the weekly data chunk that had the highest power associated with the 12-hour cycle (22-29 February 2012). The aerosol data are shown as a concentration anomaly from the seasonal mean (black). The anomaly data are broken down into the 12-hour cycle component (cyan), the 24-hour cycle component (yellow), and the combination of the 12- and 24-hour cycles (green), as computed by the power spectral analysis.**



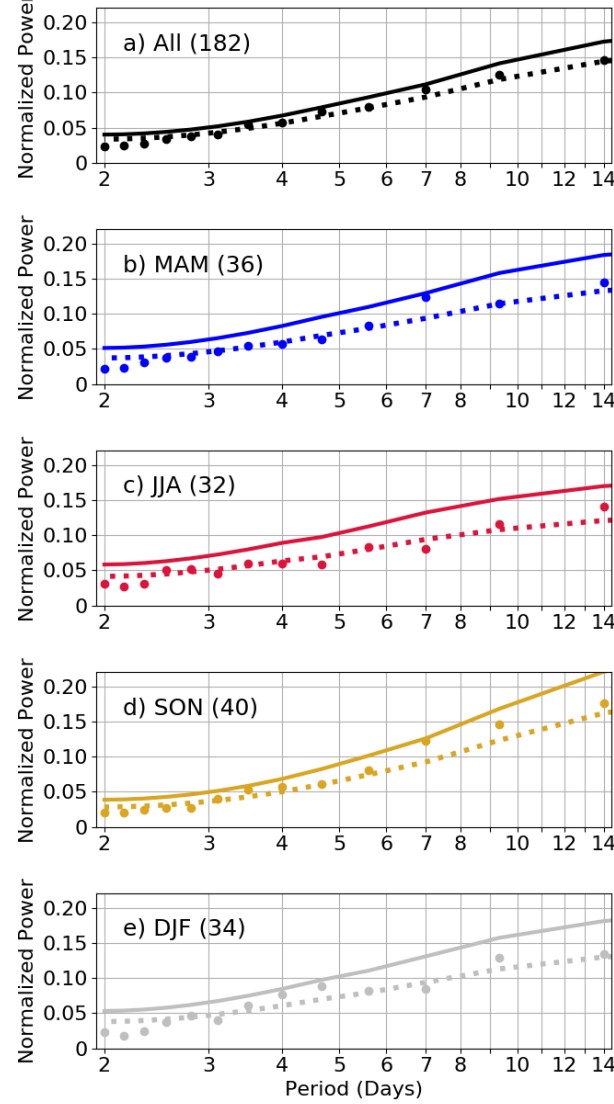

**Figure 13: Normalized power spectra for 2-14 day cycles for the total aerosol number concentrations from the CPC for the entire period (a) and by season (b-e). The dots represent power associated with the data. The dashed lines represent an estimate of the red noise power spectrum for each data set, and the solid lines represent the 99% significance testing level, as described in the text. The values in the parentheses are the number of 28-day data chunks used in this analysis.**



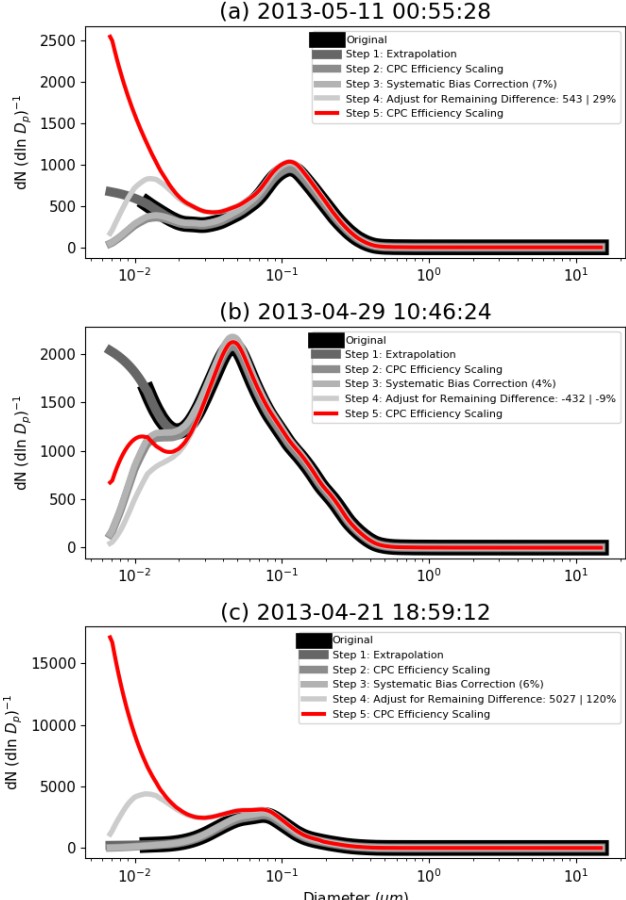

**Figure A1: Three examples of the adjustments made to the original TDMA aerosol number size distributions and the final aerosol number size distribution post-adjustments (red).**





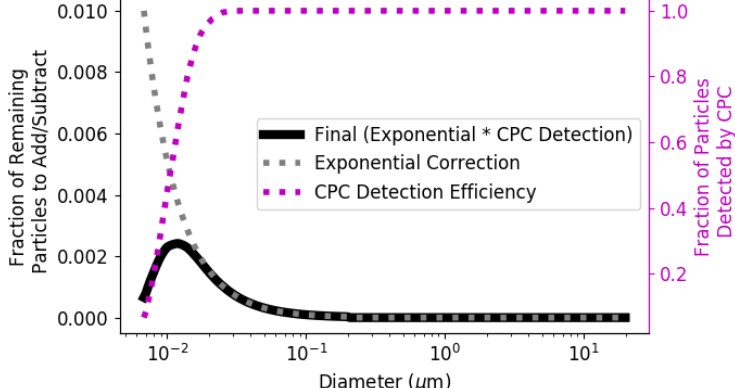

**Figure A2: Fraction of particles to either add or remove from the size distribution during Step 4 of the adjustments (black), which was based on the multiplication of an exponential function (cyan) and CPC detection efficiencies (magenta, right axis).**




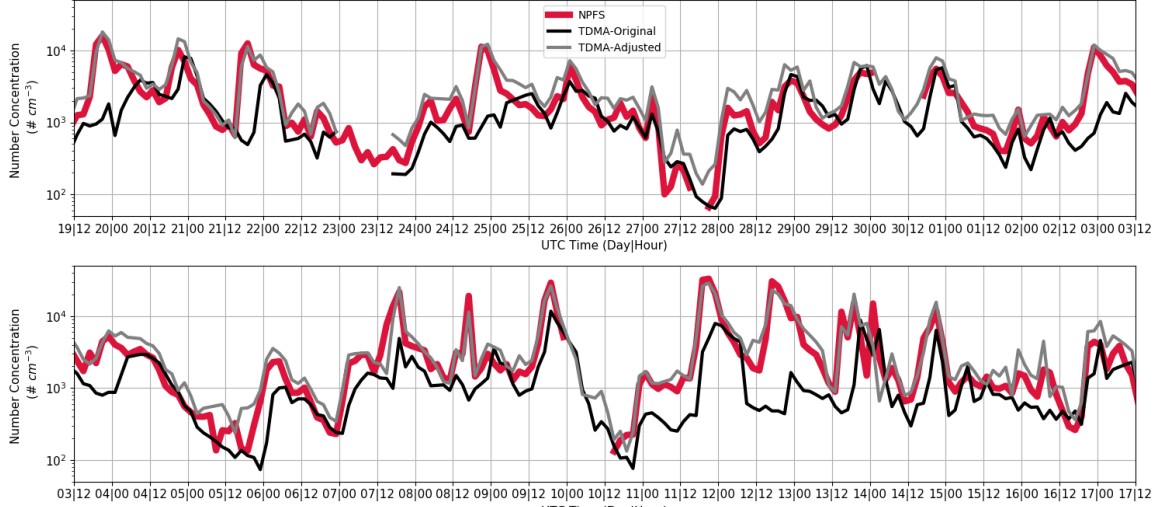

**Figure A3: Time series of the integrated aerosol number concentrations between 7 and 30 nm in the New Particle Formation Study (red) and the SMPS+APS size distributions both before the adjustments (Original, black) and after the adjustments (Adjusted, grey). The dates included were 19 Apr. 2013 through 17 May 2013.**

