# Peer review of "Quantifying aerosol size distributions and their temporal variability in the Southern Great Plains, USA"

_Atmospheric Chemistry and Physics, 2019_

## Referee Comment (RC1) · Anonymous Referee #3 · 7 Jul 2019

In their manuscript, the authors present an analysis of 5 years of data with quality-controlled data that is carefully analysed using power spectral analysis. The main finding is that for the smallest particles, there is a diurnal cycle during the whole year and not only during springtime. Also, a 12-hour cycle is found. The paper is for a large part a description of the methods of achieving the analysis, and in this it is interesting and well-written. The conclusions are somewhat light and mostly agreeing with previous results, but the finding of the continuous cycle in the smallest particles, as well as the interesting application of the statistical methodology make this paper a good addition to the literature. Overall the paper is well written and I recommend that the article is published in ACP.

[Figure]

I had a few minor comments that could be addressed before publication, which I have listed below.

* I am not sure whether replacing the season names with MAM. JJA etc. increases clarity. At least for me, it caused more confusion than just using the season names with a definition.

* page 4, line 25: "…concentrations around 3 $\mu$m was a data artifact." Do the authors have a explanation for the cause of the artefact? Is this related to the factors given at page 5, line 3 (" It is important to note…")?

* Figure 4 and corresponding text: did I understand correctly that a bolded box means that the 5-95% range is significantly different than in the ALL case? how is this determined? Not being a statistician, I do not fully understand how this is determined, and maybe an explanation could be useful for many readers too.

* Figures 6, 8 and 10: the local time could be indicated as well as the UTC time. Alternatively, the solar noon and midnight could be shown in the plot.

* page 11, line 13: "The similarities between the timing of the peak concentrations of the 12-hour cycles for NT and N7-30nm further demonstrate the regulating relationship that N7-30nm has on NT ." - What is meant with regulating relationship? The smaller particle range seems to be dominating the size distribution, and therefore the total number follows the N7-30nm , but I don't consider this as regulating. This could maybe be clarified.

* Page 13, line 16: "Because size-resolved measurements for a longer time period were unavailable, cycles in aerosol number concentrations for periods of days to weeks were tested only for NT" I did not fully understand, I thought that the whole dataset was a size-resolved dataset?

---

## Referee Comment (RC2) · Anonymous Referee #1 · 13 Jul 2019

The authors describe a quality controlled four-year dataset of aerosol size distribution with diameter ranging from 7 nm to 14 microns. The dataset was developed by combining measurements from SMPS, APS, and CPC at the DOE SGP site. Statistics of aerosol number, surface, and volume concentrations are presented for different seasons. The authors also carried out power spectral analysis of the temporal variation of aerosol size distribution and show a diurnal cycle in the concentration of small particles ranging from 7 to 30 nm for all four seasons. The diurnal variation is attributed to new particle formation.

The dataset will be useful for validating models, and future studies of aerosol pro-

cesses. The key results presented largely confirm findings of earlier studies. The topic is well suited for Atmospheric Chemistry and Physics, and overall the manuscript is well written. Following are my comments and suggestions.

(1) One focus of the manuscript is the quality-controlled aerosol size distribution dataset. Were particle losses through the inlet and inside APS (especially for coarse mode particles) taken into consideration?

(2) Equation 1- I don't think this is how lognormal aerosol size distribution is defined. Is N(ln(Dp) cumulative size distribution? If so, the limits of integration are incorrect.

(3) Page 5, Line 26: Reference Wang et al., 2009 is missing.

(4) Equation 3: please check the numerator on the righthand side.

(5) Page 9, line 12: The peak of small particle concentration occurs around UTC 22-24 (CST 16:00-18:00) during winter. I am wondering if boundary layer deepens until CST16:00-18:00 during winter time.

(6) Page 10, line 26: "The peak concentrations of the 12-hour cycle for all seasons occurred between 04 and 12 UTC (23 and 07 CDT) and between 16 and 24 UTC (11 and 19 CDT) for both N_T and N7-30nm." There is no second peak for N_T or N7-30nm between 4 and 12 UTC, at least for MAM and DJF (Figures 5 and 7).
* * *

---

## Author Comment (AC1) · 16 Aug 2019

Response to Referee Comments (RC1) for acp-2019-131
"Quantifying aerosol size distributions and their temporal variability in the Southern Great Plains, USA"
Referee Comments received on 7 July 2019

We would like to thank the reviewer for their time and comments. We have responded to their comments below.

1) I am not sure whether replacing the season names with MAM. JJA etc. increases clarity. At least for me, it caused more confusion than just using the season names with a definition.

The authors prefer to use the MAM, JJA, SON, and DJF terminology since it reminds the readers what data went into these statistics. However, we did add the following statement on P4, L15 to make a stronger connection between these acronyms and the seasons.

"Throughout this manuscript, the terms MAM, JJA, SON, and DJF can be used interchangeably with spring, summer, autumn and winter, respectively."

2) page 4, line 25: "...concentrations around 3 m was a data artifact." Do the authors have a explanation for the cause of the artefact? Is this related to the factors given at page 5, line 3 (" It is important to note...")?

The cause of the data artifact was likely related to the size of the bin widths around 3 microns in the APS data processing. We have added the following "... data artifact, which is believed to have been caused by inaccurate size bin boundaries determined from the initial instrument calibration." at the end of the current sentence to provide this information in the manuscript.

3) Figure 4 and corresponding text: did I understand correctly that a bolded box means that the 5-95% range is significantly different than in the ALL case? how is this determined? Not being a statistician, I do not fully understand how this is determined, and maybe an explanation could be useful for many readers too.

The reviewer did correctly understand that the bolded boxes in Figure 4 represented instances where the seasonal 5-95% range is significantly different than the ALL data. This determination of statistical significance was determined in the same manner as for all the statistics shown in Figure 4. We had provided an example of how the statistical difference works for the DJF mean on page 5, Lines 20-29. We have also added the following statement after the example, "The same process was completed for the median, IQR, and R595 statistics for each season." to make it clearer that the process was the same for all the statistics.

4) Figures 6, 8 and 10: the local time could be indicated as well as the UTC time. Alternatively, the solar noon and midnight could be shown in the plot.

We agree that adding local time would be helpful here. As such, we have recreated Figures 6, 8, 10, and 11 to include two axes for both UTC time and Central Daylight Time (UTC-5). We have also made this change to the corresponding figures in the Supplement.

5) page 11, line 13: "The similarities between the timing of the peak concentrations of the 12-hour cycles for NT and N7-30nm further demonstrate the regulating relationship that N7-30nm has on NT ." - What is meant with regulating relationship? The smaller particle range seems to be dominating the size distribution, and therefore the total number follows the N7-30nm , but I don't consider this as regulating. This could maybe be clarified.

We can see why the reviewer doesn't like the term regulating, as it can be interpreted in several ways in this context. We have replaced this sentence to read "The similarities between the timing of the peak concentrations of the 12-hour cycles for NT and N7-30nm further demonstrate that the variability in N7-30nm is the driving mechanism for the variability in NT. " to make our point clearer.

6) Page 13, line 16: "Because size-resolved measurements for a longer time period were unavailable, cycles in aerosol number concentrations for periods of days to weeks were tested only for NT" I did not fully understand, I thought that the whole dataset was a size-resolved dataset?

5 years (2009 through 2013) of data had size-resolved aerosol distributions. However, 5 years of data did not provide large enough seasonal samples for testing cycles with longer time periods (several-day to several-week cycles, Section 4.3). This comment in the conclusion referred to this part of the analysis. We have added some clarifying statements in the conclusion section to make this clearer.

---

## Author Comment (AC2) · 16 Aug 2019

Response to Referee Comments (RC2) for acp-2019-131
"Quantifying aerosol size distributions and their temporal variability in the Southern Great Plains, USA"
Referee Comments received on 13 July 2019

We would like to thank the reviewer for their time and comments. We have responded to their comments below.

The authors describe a quality controlled four-year dataset of aerosol size distribution with diameter ranging from 7 nm to 14 microns. The dataset was developed by combining measurements from SMPS, APS, and CPC at the DOE SGP site. Statistics of aerosol number, surface, and volume concentrations are presented for different seasons. The authors also carried out power spectral analysis of the temporal variation of aerosol size distribution and show a diurnal cycle in the concentration of small particles ranging from 7 to 30 nm for all four seasons. The diurnal variation is attributed to new particle formation.

The dataset will be useful for validating models, and future studies of aerosol processes. The key results presented largely confirm findings of earlier studies. The topic is well suited for Atmospheric Chemistry and Physics, and overall the manuscript is well written. Following are my comments and suggestions.

1) One focus of the manuscript is the quality-controlled aerosol size distribution dataset. Were particle losses through the inlet and inside APS (especially for coarse mode particles) taken into consideration?

Particle losses for the SMPS size distribution were estimated and accounted for in the SMPS data. Particle losses for the APS size distribution were not taken into consideration, but based on the authors' experiences, it is estimated that particle losses in the APS were likely small for most of the APS size distribution. We have added the following statement in the Appendix where we explain the quality control methodology.

"Here, it is important to note that estimated corrections were made to the SMPS size distributions to account for potential particle losses due to diffusion in the inlet and system tubing. Corrections were not made to the APS size distribution data for possible particle losses within the inlet and system tubing, but it is expected that these losses are likely small for most of the APS size distribution. For example, experiments have shown approximately unit transmission efficiencies for particles with diameters up to 4 µm for the SGP inlet system. For larger sizes where low particle counts make it difficult to characterize transmission efficiencies experimentally, modeled transmission efficiencies predict significantly increasing biases for particles with diameters greater than ~10 µm (Bullard et al., 2017)."

We have also added a clarifying statement in the text (Page 5, Line 4), where we mention a decrease in the transmission efficiency in the APS for the largest particles, and now explicitly state that this was not corrected for in this dataset.

Reference: Bullard, R. L., Kuang, C., Uin, J., Smith, S. and Springston, S. R.: Aerosol Inlet Characterization Experiment Report. U.S. Department of Energy ARM Climate Research Facility. DOE/SC-ARM-TR-191, doi:10.2172/1355300, 2017.

2) Equation 1- I don't think this is how lognormal aerosol size distribution is defined
Is N(ln(Dp)) cumulative size distribution? If so, the limits of integration are incorrect.

We have updated this equation to accurately reflect the aerosol number size distribution, as opposed to the aerosol number concentration within a specific size bin, which was present in the original version.

3) Page 5, Line 26: Reference Wang et al., 2009 is missing.

We have included this reference to the reference list.

4) Equation 3: please check the numerator on the righthand side.

We thank the reviewer for catching the additional power of two that showed up in the numerator of this equation. We have fixed the numerator of Equation 3.

5) Page 9, line 12: The peak of small particle concentration occurs around UTC 22-24 (CST 16:00-18:00) during winter. I am wondering if boundary layer deepens until CST16:00-18:00 during winter time.

Unfortunately, boundary layer height data at SGP for our 2009-2013 period are estimated from 4x daily (at approximately 5:30, 11:30, 17:30, and 23:30 UTC) radiosondes that cannot resolve exactly when the boundary layer height reaches its maximum depth, which is typically around 20:00-21:00 UTC in the winter time but can be later (Liu and Liang, 2010). However, we are not stating that the timing of peak boundary layer depth needs to occur simultaneously with the peak concentrations of $N_{7-30nm}$ at the surface, as it can take up to several hours to mix aerosol particles at the top of the boundary layer down to the surface, depending on their altitude and the vertical mixing time scales (e.g., Chen et al. 2018). Also, this vertical mixing process would typically be slower in wintertime boundary layers that are more stable than in the other seasons, which may also partly explain the several-hour shift in peak $N_{7-30nm}$ concentrations in the winter time. We have added some additional discussion about this vertical mixing process as well as an additional figure (see below) and a related discussion regarding the seasonal evolution of the SGP boundary layer for the 5-year focus period of this study. With these changes, we believe that we have provided additional evidence to support our statements regarding the importance of the boundary layer development for each season's $N_{7-30nm}$ diurnal cycle.

[Figure]

Figure 9: Diurnal cycle of boundary layer heights at SGP for each season, as estimated from radiosonde data. The circles represent the median boundary layer height for the top 25% of the weekly data in terms of power associated with the diurnal cycle in N7-30nm (High Power). Similarly, the diamonds represent the median boundary layer height for the bottom 25% of the weekly data (Low Power). The horizontal lines above and below the circles and diamonds represent the 25th and 75th percentiles (interquartile ranges) for this data. The numbers in parentheses represent the number of weekly time periods used in this analysis. The abscissa offset for each radiosonde launch time is for viewing purposes and does not reflect any shift in timing for each of the 4 radiosonde launch times for the different seasons.

6) Page 10, line 26: "The peak concentrations of the 12-hour cycle for all seasons occurred between 04 and 12 UTC (23 and 07 CDT) and between 16 and 24 UTC (11 and 19 CDT) for both $N_T$ and N7-30nm." There is no second peak for $N_T$ or N7-30nm between 4 and 12 UTC, at least for MAM and DJF (Figures 5 and 7).

We are unclear what the reviewer is referring to in terms of their comment that "there is no second peak for NT or N7-30nm between 4 and 12 UTC," as there is no figure that corresponds to this statement in the manuscript. In Figures 5 and 7, there is a second significant peak in the power spectrum for the 12-hour cycle, particularly for ALL, MAM, and DJF. We believe the reviewer may be referring to Figures 6 and 8 when making the above statement. However, Figures 6 and 8 are only representative of the 24-hour cycle and does not include any information about the 12-hour cycle.

Below are the same figures as Figures 6 and 8a, but for the 12-hour cycle, which pictorially demonstrates the statement highlighted by the referee above. However, since this is only a

minor point in the manuscript, we believe that including these figures in the manuscript is not necessary. We did add the phrase "(not shown)" in the manuscript after the statement to make it clearer that this statement cannot be found in any of the figures shown in the current manuscript.

[Figure]

**Figure (Above): Normalized frequency of the daily time of peak concentrations associated with the 12-hour cycle in $N_T$. This figure only includes weekly data chunks that had normalized power associated with the 12-hour cycle greater than that of the corresponding seasonal estimate of the red noise spectrum power. The numbers in parentheses represent the number of weekly data chunks that met this criterion.**

[Figure]

**Figure (Above): Normalized frequency of the daily time of peak concentrations associated with the 12-hour cycle in $N_{7-30nm}$. This figure only includes weekly data chunks that had normalized power associated with the 12-hour cycle greater than that of the corresponding seasonal estimate of the red noise spectrum power. The numbers in parentheses represent the number of weekly data chunks that met this criterion.**